# Novel Methods to Measure Height and Volume in Healthy and Degenerated Lumbar Discs in MRIs: A Reliability Assessment Study

**DOI:** 10.3390/diagnostics12061437

**Published:** 2022-06-10

**Authors:** Nadya Guellil, Neha Argawal, Magnus Krieghoff, Ingmar Kaden, Christian Hohaus, Hans-Joerg Meisel, Philipp Schenk

**Affiliations:** 1Department of Plastic and Hand Surgery, Burn Unit, BG Klinikum Bergmannstrost Halle, 06112 Halle, Germany; nadya.guellil@bergmannstrost.de; 2Department of Neurosurgery, BG Klinikum Bergmannstrost Halle, 06112 Halle, Germany; neha.argaval@bergmannstrost.com (N.A.); hj.meisel@googlemail.com (H.-J.M.); 3Institute of Radiology and Neuroradiology, BG Klinikum Bergmannstrost Halle, 06112 Halle, Germany; magnus.krieghoff@bergmannstrost.de (M.K.); ingmar.kaden@bergmannstros.de (I.K.); 4Department of Neurosurgery, Hospital Dessau, 06847 Dessau, Germany; hohauschristian@gmail.com; 5Department of Science and Research, BG Klinikum Bergmannstrost Halle, 06112 Halle, Germany

**Keywords:** degenerated lumbar disc, disc height, disc volume, height measurement, volume measurement, intra- and inter-observer reliability

## Abstract

Background: In the regeneration and therapy of degenerated intervertebral discs, the height, volume or categorizing assessments, such as Pfirrmann classification, are used to quantify the discs themselves and the effects of therapy. Here, the question of transferability, in the sense of reliability, of the results arises in the common exchange. Methods: We have investigated two established and a newly developed (9-point measurement), easy to use methods for height measurement and volume measurement on degenerated and healthy lumbar intervertebral discs of 66 patients regarding inter- and intra-observer reliability. Results: In overview, we found very different reliabilities. While the intra-observer reliability showed good to excellent agreement for both healthy and degenerated lumbar discs for the height and volume measurements, the inter-observer reliability was low or moderate in some cases. The 9-point method for height determination consistently showed better reliability for both healthy and degenerated discs, for both intra- and inter-observer reliability, compared to the two established methods. Conclusions: We recommend using the 9-point measurement as the method to communicate lumbar disc height, both for healthy and degenerated discs. Due to the partly low or moderate reliability, significant differences in the measured heights can already occur, which can lead to a worsened comparability.

## 1. Introduction

Ever since magnetic resonance imaging examinations have been used to diagnose pathologies of the vertebrae, the objective and quantitative measurement of intervertebral disc degeneration has been a subject of scientific research. Reliable methods for objective quantification of disc degeneration are the mandatory basis of well-designed studies and are a potential foundation to evaluate treatment of degenerated discs. Vertebral disc degeneration is a complex multifactorial process constituting of cellular and biomechanical alterations in the disc tissue composition [1,2], caused notably by the mechanical overloading, oxidative damage to endplate chondrocytes, catabolic cell response and decreased water binding capacity of the extracellular cell matrix [3]. Previous studies have observed a correlation between water binding capacity of the disc and its histopathological structure [4,5,6].

To assess disc degeneration, the qualitative measures of disc morphology in MRI scans, T2 signal intensity, Modic changes, disc height measurement, endplate shape and the Pfirrmann classification are generally recognized [7,8,9,10,11,12]. The main drawback of subjective assessment categorizing discs into different levels of degeneration has low reliability. Even using healthy discs as reference, e.g., to examine and compare signal intensity of degenerated discs, cannot eliminate inter- and intra-observer differences [7,11]. Using a categorical classification system like the Pfirrmann classification only reaches moderate to good reliability and an interrater agreement of 83% [7,8,9,13]. Additionally, patho-morphological, biochemical methods of disc degeneration measurement have been established but non-invasive methods have a greater potential of clinical implications [14,15].

Neubert et al. compared a semi-automated method with a conventional manual method to measure disc height and volume from MR images. The study showed strong reproducibility for both the methods and very strong intra-class correlation for disc height and volume [16]. The approach to measure disc height and volume is based on the fact that reduced water content, bulge [8], protrusion or extrusion affect the height and volume of the vertebral disc [17,18,19]. Due to the specific three-dimensional configuration of the vertebral disc, which differs depending on its location and degenerative state, the task is more complicated than often presented [20].

Some researchers support the opinion that measurement methods introduced in conventional X-rays can be transferred to MRIs [19]. It is often not completely depicted where and how exactly a disc height is or should be measured in MRI scans. Mostly it is measured in the disc’s “center” [21] or at the highest distance in the center of the disc in sagittal layers of T1-weighted images [8,22,23]. Another approach to measure disc height is the use of an arithmetic mean of the anterior and posterior height of the median sagittal layer [11,18,19,22,24,25] or of different defined heights of the disc in sagittal layers [16,26]. Other authors describe a mean disc height, obtained by dividing the sagittal disc area with the disc’s diameter [27]. To evaluate disc height variations, the heights have been estimated between two points at the center of the disc’s two adjacent vertebral points.

The variety of introduced measurement methods makes applicability and transferability questionable. Videman et al. indicated limited sensitivity of MRI parameters such as using disc height reduction as a degenerative marker and postulated low clinical significance [28]. To define a disc volume of a degenerated disc using MRI scans, different approximation methods have been introduced. Neubert et al. estimated disc volume by multiplying the sum of the disc areas in sagittal layers with the thickness of the layers [16]. Other authors used semi-automated or interpolation methods to estimate the disc volume [25,29]. Pfirrmann et al. introduced a system based on the Cavalieri method, in which first the disc area in sagittal layers is measured and second the disc volume is obtained by multiplying the sum of the disc areas with the layer thickness [22].

Besides manual measures, technological advances have introduced new semi-automated and automated measurement methods to objectively assess and quantify disc degeneration [16,30,31,32]. In the current literature, the variability and poor methodical description of quantitative measures of disc height and volume make reproducibility and reliability difficult and often lack adequate description of applied statistical methods. To examine the quality of newly introduced automated measurement methods, a comparison to established manual methods is needed. 

Since quantitative measures should theoretically grant more reliable results, this study aims to propose a novel, simple and manual method to estimate disc height and volume in lumbar discs and compare it with commonly used methods (H1: disc height measured in the center of the midsagittal layer; H2: arithmetic mean of anterior and posterior disc height in midsagittal layer). Furthermore, we investigate the reliability and inter-observer assessment of the quantitative measures of disc height and volume.

## 2. Materials and Methods

Data of patients who had a mono-segmental lumbar disc herniation at L4/5 or L5/S1 and underwent primary sequestrectomy were analyzed retrospectively. The evaluated patients were part of a study evaluating the efficacy of autologous disc-derived chondrocyte transplantation. Their magnetic resonance imaging (MRI) records were randomly picked three months to five years after the surgery. The MRI sequences were carried out from 2002 to 2008 with a 1.5T MRI scanner (scanner: Gyroscan NT Intera, Philips Medical Systems, software version: NT 8.1.1\1.3) using the same examination protocol. All patients were examined using native sagittal angled T1- and T2-weighted sequences and T1-weighted sequences angled on the herniated disc in 4 mm thick layers (T2 sagittal: TE 120ms, TR 3274ms; T1 sagittal: TE 12ms, TR 550ms; T2 axial: TE 110ms, TR 2322ms; T1 axial: TE 18ms, TR 630ms; picture reconstruction matrix: 512 × 512 pixel).

The MRI scans were analyzed using the PACS-software (IMPAX, Agfa HealthCare). Using the software, different sequences can be displayed next to each other and correlating measurement points can be indicated. The picture management tools magnifying glass, windowing tool, measurement ruler and molding tool were used. 

The scans were analyzed independently by two observers, a surgeon and a radiologist. The assessment of the measures followed a strict protocol. The first observer (surgeon, NG) examined the scans twice. The measures are described in the following as observation 1, 2 and 3, where observation 1 and 2 represent the first and second measurements of the first observer with a time delay of 4 weeks and observation 3 represents the measurement of the second observer (radiologist, MK). 

In T2-weighted sequences, the Pfirrmann classification of the herniated disc and of an adjacent healthy disc at level L3/4 were determined to define the stage of degeneration of the disc [8,9]. Patients who received an operative sequestrectomy due to a herniated disc at level L4/5 or L5/S1 and had a healthy disc at level L3/L4 (Pfirrmann 1 or 2) were included in the study.

Measurement techniques for disc height and volume:

Three different heights of the lumbar discs, respective height measurement methods, denoted as H1, H2 and H3 have been used. Height H1 of the discs was measured in the center (visual center between the anterior and posterior intervertebral disc space) in the midsagittal plane (Figure 1). 

Height H2 was calculated as the mean of the anterior (ventral) and posterior (dorsal) measured disc height in the midsagittal plane. Height H3 was obtained by calculating the mean of nine measuring points (including the measuring points of H1 and H2). The nine measuring points were measured at the midsagittal and two parasagittal sequences running through the pedicels of the vertebrae (see Figure 2). To ensure a better reliability of the method, the following aspects were to be followed during a measurement of H1–H9. Figure 1 follows these rules: The endplates were not included. The caudal vertebrae formed the basis of the disc height at the posterior and anterior margin. Potential osteophytes were excluded. When in doubt of which part of the vertebral body was an exophyte and which was the bony endplate, an imaginary line was drawn parallel to the vertebral body contour. Bulges or protrusions of the disc should not be included in the measurement. To be certain of measuring the intervertebral disc space between the vertebral bone, the visual presentation of the vertebrae and disc can be taken into account in axial planes. In each sequence, anterior, central and posterior heights were recorded. The lines drawn to examine heights H1, H2 and H3 were perpendicular to the adjacent cover plate.

The disc volume vas calculated by multiplying the sum of the measured disc areas in every sagittal MRI layer with the scan thickness of 4mm, as recommended from Pfirrmann et al. 2006 [22], see Figure 3. The measurement of disc height and volume was always performed in the same way. Bulging or protrusions of the disc, as well as the periosteum and osteophytes of the adjacent vertebrae, were not included in the measurements.

Statistical analysis:

To investigate differences between the three observations and the three height measurement methods, general linear model (GLM) for repeated measures (rm) was used. The observation was used as between subject factor and the height measurement methods as within subject factor. To check differences in disc volume between observations, a GLM was used as well, with the observation as between subject factors. If the Mauchly test of sphericity showed significance, the results of the Greenhouse–Geisser test were given. Post hoc, the Bonferroni test was used for pairwise comparison. The healthy and degenerated discs were analyzed separately. Descriptive values are presented as mean and 95% confidence interval. For the height and volume measurement methods, the intra- and inter-observer reliabilities were calculated using intra-class correlation coefficient for absolute agreement and single measures (ICC(3, 1)). The ICC can reach values between 0 and 1, where values greater than 0.9 stand for perfect agreement. Kendall’s tau was used to calculate the intra- and inter-observer reliability for the Pfirrmann classification. Kendall’s tau values range from 1 for identical observations to –1 for completely different observations. If Kendall’s tau is 0, no agreement is present. SPSS V27 (IBM Corp. Released 2020. IBM SPSS Statistics for Windows, Armonk, NY, USA: IBM Corp.) was used for statistical analysis.

## 3. Results

In total, datasets of 60 patients (34 men, 26 women) with a mean age of 36 ± 10 years were analyzed in this study. Eleven patients were excluded due to unhealthy discs at level L3/4 with a Pfirrmann grade higher than 2 or because of incomplete data sets. Ten patients showed a degenerated disc at level L4/L5 and 50 at level L5/S1. The measured heights H1, H2 and H3 and volumes for each observation are given in Table 1.

For the heights of the healthy discs, the general linear model (GLM) showed differences between the three height measurement methods (*p* < 0.001) and the three observations (*p* < 0.001). All post hoc pairwise comparisons performed showed significance with *p* < 0.001. Each of the height measurement methods measured a different disc height. Averagely in the observations of healthy discs, the H2 method resulted in the smallest disc height with 5.7 mm (5.4–5.9) while the H1 method quantified the largest disc height with 9.1 mm (8.7–9.4). The nine-point measuring method H3 led to healthy disc heights of 6.5 mm (6.2–6.7), which were in between the heights of the H1 and H3 methods. Compared to the H3 method, the overall healthy disc height measured with H1 was 2.7 mm (40%) greater and with H2, 0.8 mm (13%) lower for healthy disc heights.

The post hoc pairwise comparison of the observations showed significant differences in healthy disc heights (*p* < 0.001). In general, with 6.6 mm (6.2–6.9), observation 3 showed the lowest mean heights in healthy discs in all methods compared to observation 1 (7.1 mm (6.9–7.4)) and observation 2 (7.5 mm (7.2–7.8)). Observation 1 and 2, although significantly different, resulted in a similar mean healthy disc height in all observations. The mean difference between observation 2 and 3 was 0.9 mm (14%).

Disc volume measurements were significantly different between each observation for healthy discs (*p* < 0.001). The mean difference between observation 1 and 2 was 7%. Observation 1 and 2, conducted by the same observer, showed a mean difference of 21% and 15% compared to observation 3. 

For the degenerated disc heights, significant effects for the different height measurement methods (*p* < 0.001) and the observations (*p* < 0.001) were found. Each post hoc pairwise comparison showed significance for both observers and methods, with *p* < 0.001. The method H1 showed an overall mean disc height of 5.9 mm (5.4–6.4), which is 1.1 mm (23%) greater compared to H2, with 4.8 mm (4.4–5.2) and 1.5 mm (34%) greater compared to H3 with 4.4 mm (4.0–4.7). The mean degenerated disc height between H2 and H3 differed by 0.4 mm (9%). 

Observation 3 showed the lowest degenerated disc heights with 4.5 mm (4.1–4.9). As seen by the values of healthy discs, the mean degenerated disc heights of observation 1 (5.1 mm (4.7–5.5)) and 2 (5.4 mm (5.0–5.9)) showed a small difference of 0.3 mm (6%), being significantly different. The mean disc height in observation 3 was 0.6 mm (12%) smaller than in observation 1 and 0.9 mm (17%) smaller than in observation 2. The mean disc height of observation 1 and 2 differed by just the measurements of the degenerated disc volumes, which showed 21% and 15% higher volumes in observation 2, compared to observation 1 and 3. The measured disc volume between observer 1 (observation 1) and observer 2 (observation 3) did not differ significantly (*p* = 0.237).

The results of the intra- and inter-observer reliability are given in Table 2. Both, in the healthy and the degenerated disc, the nine-point measurement method H3 showed the best intra-observer reliability compared to H1 and H2 methods. The ICC for the height of degenerated discs determined with H3 reaches excellent values (0.910), while the ICC for healthy discs measured with H2 reached lower (moderate agreement) values (0.677). The intra-observer reliability for the volume measures at healthy and degenerated discs are also very good with 0.887 and 0.913, respectively.

The inter-observer reliability reached lower values for reliability than the intra-observer reliability. The H3 method showed the best ICC values at all considered comparisons. The inter-observer reliability of the volume measurement method is slightly lower than the intra-observer reliability, with ICC values of 0.701 and 0.860, which testifies excellent agreement, respectively [33].

The herniated discs treated by sequestrectomy were considered as degenerated and showed Pfirrmann classification from grade II to V. Only in observation 1, Pfirrmann grade II could be found. The most frequent grades were III and IV. In Table 3 the frequencies of Pfirrmann classification grades in the different observations are listed.

The reliability (Kendall’s tau) for the Pfirrmann classification is given in Table 4. Both, intra- and inter-observer reliability show low to moderate agreement.

## 4. Discussion

The aim of this study was to investigate the reliability of methods to measure disc height and volume. The methods investigated here represent simple measurement techniques in MRI images, which can be performed by simple means, with most of the available MRI software. Evaluating vertebral disc height and volume in MRIs plays an important role in reaching comparability and interpretability in scientific studies investigating lumbar vertebral disc pathology. Disc height and volume offer a qualitative numeric parameter to evaluate degeneration [7,8,13,34,35,36,37,38,39].

This study introduced the nine-point height measuring method H3 and a volume estimation method in MRIs. The reliability for repeated intra- and inter-observer measurements was good to very good. We showed that the height and volume measurement of healthy discs had lower inter-observer reliability compared to those of degenerated discs. The Pfirrmann classification to categorize disc degeneration has shown good reliability in previously published studies and, thus, is commonly used [9,13,40]. In this study, however, it only reaches low to moderate values for reliability, which can be explained by subjective factors such as loss of signal intensity, distinguishability of nucleus and annulus, loss of disc height and disc collapse. In addition, the grading of disc degeneration in an MRI might be classified by instinct and not objectively [8,9,12,41].

Videman et al. have found that the sensitivity of the loss of disc height as indication of disc degeneration is poor [28]. When put into practice, measuring disc height as one value in the midsagittal center of the disc (H1) lacks precision. Whether it is measured at the optical or mathematical center of the disc or at the highest point at the center of the disc is not clear, and results in substantial difference in the determined disc height. Only relying on this inconsistent parameter makes it prone to incorrect values. Measuring height as a mean of the anterior and posterior midsagittal height (H2) compensates this assumed source of error by using a mean of two values. Theoretically, nine points would compensate anatomical irregularities of the disc’s body, which are not considered by these methods. When comparing the three height measuring methods, H1 seems to overestimate the disc height up to 40%, while H2 underestimates it, compared to H3. Following this finding, the assumption can be made, that the 9-point-method H3 might be more representative for estimating the disc height in MRIs.

Many authors state that the disc height should always be measured in the disc center [8]. This might be justified by the consideration that the outer (lateral) annulus regions are more often influenced by osteochondrosis or irregular degeneration due to scoliosis. Therefore, in the proposed H3 method, measurements from lateral layers were not included. While this study proves a very good reliability between observers for disc height measurements in general, the 9-point method seems to be the best option to quantify disc degeneration in MRIs objectively and reliably. 

The time effort to determine the disc volume is higher than using any of the height measurement methods. Considering the complex form of the lumbar vertebral disc, determining disc volume is equally important as disc height estimation to assess changes in a disc extracellular matrix and the associated water binding capacity. Since the reliability of the presented method to measure disc volume is good to very good, it represents an alternative method. 

When evaluating confidence in estimates, we found out that one of the main error sources in the disc volume measurement process was whether a marginal MRI slice was included or not. The comprehension of disc anatomical structures, as well as a conscious execution of the measurements, seem to affect the reliability of the method directly. The measurements of observer 1 (observation 1 and 2) had a smaller deviation than those compared to observer 2, which indicates better intra-observer reliability than the inter-observer reliability, and is further supported by corresponding intra-observer reliability analysis.

Whether the volume or height assessment as measured by the 9-point method correlates with disc degeneration still needs to be examined in future studies. Since the required time to estimate the disc height and volume with the presented methods is rather high, the implication of these findings in the recently introduced automated methods seems a practical idea [16,30]. Neubert et al. compared semi-automated methods for disc height measurement with manual assessment methods and concluded comparable reliability for both methods. When presenting new computerized methods, this study offers a basis to compare them to manual methods [16].

## Figures and Tables

**Figure 1 diagnostics-12-01437-f001:**
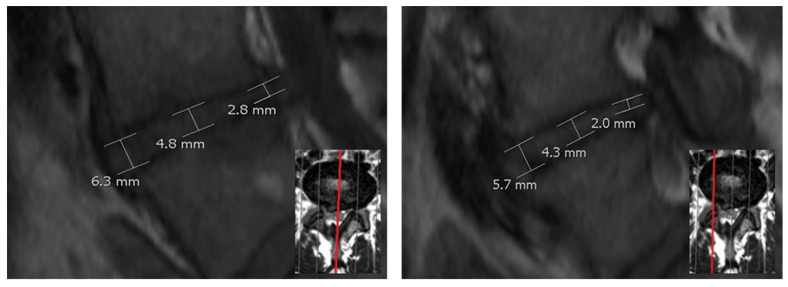
Left image: Measurement of disc height for the methods H1 (disc height in the center) and H2 (mean of anterior and posterior disc height) midsagittal (red line). Right image: Example of the H3 method using additional parasagittal (red line) layers.

**Figure 2 diagnostics-12-01437-f002:**
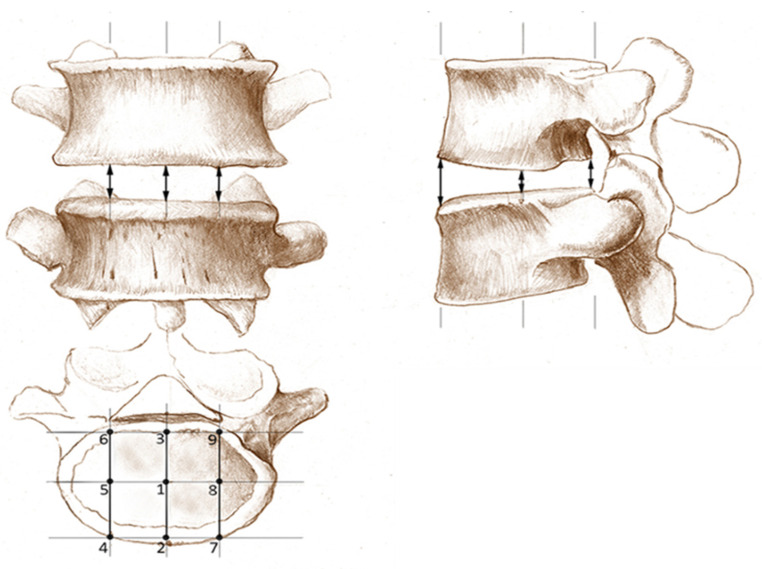
Schematic depiction of the height measurement method using the 9-point method H3. Mean heights are calculated as an arithmetic mean of the heights 1 to 9. Height H1 is measured at point 1 and height H2 is the arithmetic mean value of the points 2 and 3.

**Figure 3 diagnostics-12-01437-f003:**
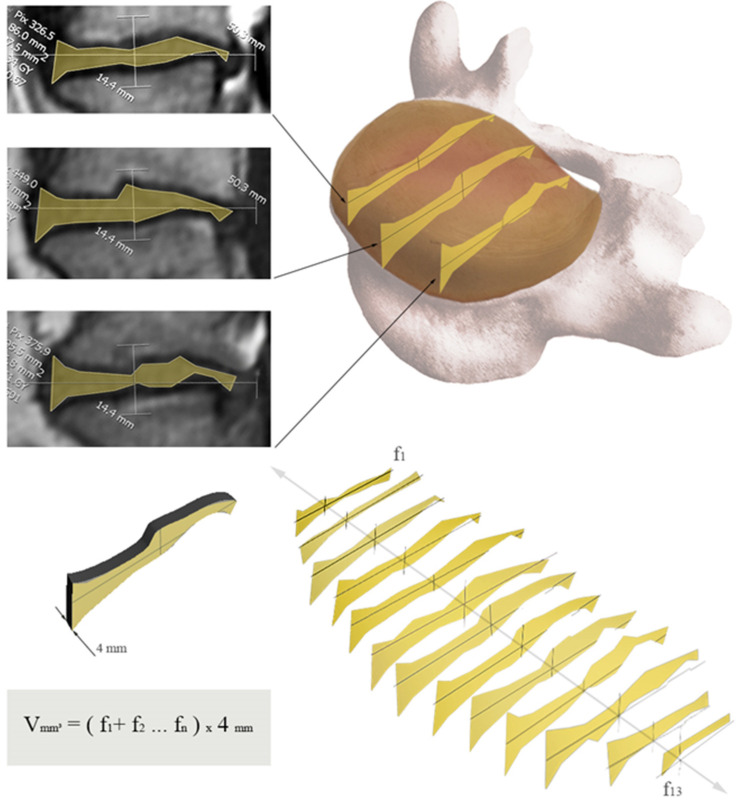
Schematic depiction of the disc volume measurement. The disc areas are measured in all sagittal layers of the MRI. To obtain a volume, the areas are summed and multiplied by the layer thickness of the scans (4 mm).

**Table 1 diagnostics-12-01437-t001:** Measured disc heights and volumes of the healthy and degenerated lumbar disc. Values are displayed as the mean, and in brackets, the lower and upper limits of the 95% confidence interval. H1: disc height in the center of the midsagittal plane; H2: mean disc height of the ventral and dorsal measured disc height in the midsagittal plane; H3: mean of nine measuring points. Observation 1 and 2 are the results from observer 1 at two different time points. Observation 3 was performed by a second observer.

			Observation 1	Observation 2	Observation 3
healthy	height [mm]	H1	9.3 (9.0–9.7)	9.5 (9.1–9.9)	8.4 (7.9–8.9)
		H2	5.6 (5.3–5.9)	6.1 (5.8–6.4)	5.3 (4.9–5.6)
		H3	6.5 (6.2–6.7)	7.0 (6.7–7.3)	6.0 (5.8–6.3)
	volume [cm³]		11.9 (11.1–12.7)	11.1 (10.2–11.9)	9.4 (8.7–10.2)
degenerated	height [mm]	H1	6.2 (5.7–6.7)	6.3 (5.8–6.8)	5.2 (4.7–5.7)
		H2	4.7 (4.3–5.2)	5.3 (4.9–5.8)	4.3 (3.9–4.7)
		H3	4.4 (4.0–4.7)	4.7 (4.3–5.1)	4.1 (3.7–4.4)
	volume [cm³]		5.6 (5.0–6.3)	6.5 (5.8–7.2)	5.3 (4.7–6.0)

**Table 2 diagnostics-12-01437-t002:** Inter- and intra-observer reliability of the methods for the measurement of disc height by three different methods (H1, H2 and H3) and the volume of healthy and degenerated lumbar discs. ICC values are given as mean, and in brackets, the lower and upper limits of the 95% confidence interval.

		Observation	H1	H2	H3	Volume
Intra-observer	Healthy		0.815 (0.710–0.885)	0.677 (0.328–0.834)	0.822 (0.246–0.935)	0.887 (0.734–0.944)
	Degenerated		0.874 (0.799–0.923)	0.842 (0.560–0.929)	0.910 (0.736–0.959)	0.913 (0.450–0.971)
Inter-observer	Healthy	1 and 3	0.500 (0.199–0.696)	0.508 (0.296–0.673)	0.634 (0.396–0.781)	0.701 (0.064–0.914)
		2 and 3	0.598 (0.159–0.799)	0.396 (0.082–0.621)	0.533 (0.032–0.788)	0.728 (0.212–0.885)
	Degenerated	1 and 3	0.725 (0.295–0.875)	0.758 (0.602–0.854)	0.792 (0.664–0.873)	0.860 (0.776–0.914)
		2 and 3	0.679 (0.233–0.850)	0.630 (0.145–0.826)	0.727 (0.358–0.869)	0.791 (0.365–0.911)

**Table 3 diagnostics-12-01437-t003:** Pfirrmann classification of degenerated discs in the investigated patients, determined by two observers. Observation 1 and 2 were performed at different time points by one observer with a time delay of 4 weeks, observation 3 were performed by a second observer.

Pfirrmann Classification	Observation 1	Observation 2	Observation 3
II	16	-	-
III	17	22	16
IV	20	27	32
V	7	11	12

**Table 4 diagnostics-12-01437-t004:** Intra- and inter-observer reliability (Kendall’s tau) for the Pfirrmann classification of degenerated lumbar disc.

	Observation	Kendall‘s Tau
Intra-observer reliability	1 and 2	0.628
Inter-observer reliability	1 and 3	0.457
	2 and 3	0.619

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
