# Peer review of "Novel Methods to Measure Height and Volume in Healthy and Degenerated Lumbar Discs in MRIs: A Reliability Assessment Study"

_diagnostics, 2022, doi:10.3390/diagnostics12061437_

Round 1

Reviewer 1 Report

The reviewer pays homage to the author's great efforts in measurement. However, the content of this paper is merely a study of measurement methods and does not suggest whether it is clinically meaningful. The reviewer cannot agree with publication.

Others

  1. Line 209: ・・・・just The measurements of ・・‥→just the measurements of

Reviewer 2 Report

General comments

This manuscript aims at proposing a novel simple and manually method to estimate disc height and volume in lumbar discs and compare it with two commonly used methods. Apart from some minor issues detailed below, MS “cleanly” depicts a novel methodology with useful and commendable application. Authors manage sufficiently to fulfill their aim.

Minor comments

(line 17) The q Here the question of transferability… ?!

(l132) Please, do not start sentences with acronyms;

(l213) … in Table 2.

Reviewer 3 Report

The manuscript entitled 'Novel methods to measure height and volume in healthy and degenerated lumbar discs in MRIs: A reliability assessment study' conducted the MRI study for evaluation of lumbar disc height and volume.

The manuscript is well-written; however, the reviewer has several concern on this manuscript.

  1. Method section (line 133) : For determination of nine measurement point,  how to determine the two parasagittal planes? Is there any differences in determining the planes dependent on the observer? 
  2. Method section (line 135): How to determine the anterior and posterior margin of disc for evaluation of disc height? This should be described in detail.
  3. Method section (line 141): How to determine the margin of disc area? This should be described in detail. Did the endplates were included or excluded? 

Round 2

Reviewer 1 Report

The reviewer entirely agrees with this disc volume measurement for methodology. However, at least for now, this method of measurement is of little use in clinical and research use, the reviewer considers. A truly useful measurement must meet some criteria.

  1. Reliability

The authors evaluate intra-observer reliability and inter-observer reliability as an evaluation of reliability. As authors mention, intra-observer reliability was acceptable, but inter-observer reliability was low. This means that this measurement method is not always useful for clinical or research use.

  1. Validity

There is no validation study. What is the relationship between the disc volume and the more accurate disc volume by this measurement method? And what is the clinical or research meaning of the disc volume by this measurement method? Or what does it correlate with clinically? There is no description regarding these issues. The reviewer does not understand the usefulness of this measurement method.

In line 247, authors describe “The aim of this study was to investigate the reliability of methods to measure disc height and volume.”, however, this measurement method is truly reliable or not, the reviewer cannot judge from this manuscript at least. The reviewer believes that at a minimum, a correlation should be shown between the disc volume measured by the method considered to be more detailed and the disc volume measured by the method.

Reviewer 3 Report

The author appropriately responded to the reviewer's comment.

Author Response

Comments and Suggestions for Authors

The author appropriately responded to the reviewer's comment.

Response to Reviewer 2:

Thank you very much for your hints to improve our manuscript.